# Comparison of the Bioactive Properties of Human and Bovine Hemoglobin Hydrolysates Obtained by Enzymatic Hydrolysis: Antimicrobial and Antioxidant Potential of the Active Peptide α137-141

**DOI:** 10.3390/ijms241713055

**Published:** 2023-08-22

**Authors:** Ahlam Outman, Barbara Deracinois, Christophe Flahaut, Mira Abou Diab, Jihen Dhaouefi, Bernard Gressier, Bruno Eto, Naïma Nedjar

**Affiliations:** 1UMR Transfrontalière BioEcoAgro N_1158, Institut Charles Viollette, National Research Institute for Agriculture, Food and the Environment, Université Lille, F-59000 Lille, France; ahlam.outman.etu@univ-lille.fr (A.O.); barbara.deracinois@univ-lille.fr (B.D.); christophe.flahaut@univ-artois.fr (C.F.); mira.abou-diab.1@ulaval.ca (M.A.D.); jihen.dhaouafi.etu@univ-lille.fr (J.D.); naima.nedjar@univ-lille.fr (N.N.); 2Laboratoire TBC, Laboratory of Pharmacology, Pharmacokinetics, and Clinical Pharmacy, Faculty of Pharmaceutical and Biological Sciences, University of Lille, 3, rue du ProfesseurLaguesse, B.P. 83, F-59000 Lille, France; 3Laboratory of Pharmacology, Pharmacokinetics and Clinical Pharmacy, Faculty of Pharmaceutical and Biological Sciences, University of Lille, 3, rue du Professeur Laguesse, B.P. 83, F-59000 Lille, France; gressier.bernard@univ-lille.fr

**Keywords:** human hemoglobin, bovine hemoglobin, bioactive peptides, antibacterial activity, antioxidant activity

## Abstract

This study focuses on the enzymatic hydrolysis of hemoglobin, the main component of cruor that gives blood its red color in mammals. The antibacterial and antioxidant potentials of human hemoglobin hydrolysates were evaluated in comparison to bovine hemoglobin. The results showed strong antimicrobial activity of the peptide hydrolysates against six bacterial strains, independent of the initial substrate concentration level. The hydrolysates also showed strong antioxidant activity, as measured by four different tests. In addition, the antimicrobial and antioxidant activities of the human and bovine hemoglobin hydrolysates showed little or no significant difference, with only the concentration level being the determining factor in their activity. The results of the mass spectrometry study showed the presence of a number of bioactive peptides, the majority of which have characteristics similar to those mentioned in the literature. New bioactive peptides were also identified in human hemoglobin, such as the antibacterial peptides PTTKTYFPHF (α37-46), FPTTKTYFPH (α36-45), TSKYR (α137-141), and STVLTSKYR (α133-141), as well as the antioxidant TSKYR (α137-141). According to these findings, human hemoglobin represents a promising source of bioactive peptides beneficial to the food or pharmaceutical industries.

## 1. Introduction

For many years, agrifood proteins have been considered an important source of biologically active peptides. Indeed, they have been widely used in the food and pharmaceutical industries as components of functional foods for humans and animals. Among these proteins, hemoglobin is of particular interest as it constitutes more than half of the proteins listed in databases listing the various activities of active peptides [1,2]. This abundance of activities gives hemoglobin a promising interest in the research and development of drugs of natural origin, offering an alternative to synthetic compounds [1,2,3].

This latter consists of a tetramer comprising two α- and two β-globin chains, each containing a proto-heme group [4].

The first hemoglobin-derived peptide with biological use was discovered in 1971 by Schally et al. [5]. It is a decapeptide that has been identified to be a 1–10 fragment of the β-chain of hemoglobin and has growth hormone-releasing action from the porcine hypothalamus. Since this discovery, other hemoglobin-derived peptides with different biological functions have been identified, such as the analgesic peptides kyotorphin [6] and neokyotorphin [7,8], corresponding to C-terminal sequences 140–141 and 137–141 of α-globin, which were discovered in 1979 and 1982 from bovine brain. Proteolytic treatment of hemoglobin also produces a series of peptides with opioid-like activity, called hemorphins, which were detected by Brantl et al. [9]. Since then, systematic studies of the peptide composition of various tissue extracts have led to the discovery of a large number of new peptides, the structure of which is mainly derived from α- and β-globin chains [10].

In particular, the hydrolysis of bovine hemoglobin by pepsin generates many peptides with different biological activities, including opioids [11,12], hematopoietic [13], and antihypertensive [14]. However, antimicrobial activity is the most commonly reported in studies [3,15,16,17]. The peptide α137-141 (neokyotorphin, 653 Da, pI of 10.5), an antimicrobial peptide found in bovine hemoglobin, has recently been identified as an effective natural preservative to protect meat during storage and distribution [3]. Both human hemoglobin and bovine hemoglobin offer the potential to produce bioactive peptides. Some studies have thus shown that human hemoglobin is an ideal substrate for proteolysis and the generation of bioactive peptides with human or animal health benefits [3]. Furthermore, by mapping specific regions of the hemoglobin molecule using synthetic peptides, Parish et al. [18] demonstrated the antimicrobial activity of intact human hemoglobin tetramers, separated alpha and beta subunits, and identified the 30 carboxy-terminal amino acids of the beta subunit as being responsible for significant antimicrobial activity [19].

The main objective of this study was to produce bioactive peptides through the enzymatic hydrolysis of human and bovine hemoglobin (used as a control group, laboratory model) and to demonstrate their bioactive properties. The focus was on the production of a specific peptide, α137-141, which is often obtained through this method [14,16,20]. The secondary objectives were to evaluate the biological activity of the peptides. To achieve this, the study measured the antibacterial activity of the hydrolysates using the agar diffusion method and determined the minimum inhibitory concentration against six different bacteria. In addition, the antioxidant activity was evaluated using several methods, such as the inhibition activity of β-carotene bleaching, the DPPH radical scavenging activity assay, the total antioxidant capacity, and the ABTS radical scavenging assay. Finally, the study identified the bioactive peptides generated by UPLC-MS/MS.

## 2. Results

### 2.1. Biological Activities of the Human and Bovine Hydrolysates

#### 2.1.1. Antibacterial Activity

##### AntimicrobialTest on Petri Dish

According to Table 1, all human and bovine hemoglobin hydrolysates showed antibacterial activity against the six bacterial species tested, with a clear zone of bacterial growth inhibition observed at a concentration of 20 mg/mL. The results showed that concentration was a key factor, irrespective of the amount of hemoglobin substrate used (1%, 2%, 8%, or 10% *w*/*v*). Overall, there was no significant difference in antibacterial activity between human and bovine hemoglobin hydrolysates against any given species, with the exception of the *Salmonella Newport* strain at all concentrations. In addition, the α137-141 peptide also showed antibacterial activity against all tested strains, confirming the results previously reported by Catiau et al. [21]. The results obtained for bovine and human hemoglobin hydrolysates agree with the results of previous research conducted by Nedjar-Arroume et al. [16,22,23].

Figure 1 shows the antimicrobial tests carried out on Petri dishes, providing a visual representation of previous results and demonstrating the antibacterial efficacy of human hemoglobin hydrolysates (1%, 2%, 8% and 10% (*w*/*v*)) obtained by treatment with pepsin.

##### MIC Determination of Antibacterial Hemoglobin Hydrolysates

According to the experimental design, the samples were evaluated for their antimicrobial activity as a function of the initial hemoglobin concentration, namely 1%, 2%, 8%, and 10% (*w*/*v*) for human and bovine hydrolysates with a degree of hydrolysis DH = 10%. The MIC values of the hydrolysates were measured, and the results showed that each sample exhibited activity against both Gram-positive and Gram-negative bacteria. Peptides active against *K. rhizophila*, *S. aureus,* and *L. monocytogenes* showed significantly lower MIC values (between 0.15 and 0.62 mg/mL) than those found for *M. luteus*, *E. coli,* and *S. Newport* (between 5 and 10 mg/mL).

The antibacterial activity of human and bovine hydrolysates against *K. rhizophila* revealed MIC values of 0.15 mg/mL for all concentrations tested (1%, 2%, 8%, and 10% *w*/*v*). Similarly, all hydrolysates, whether bovine or human, showed strong antibacterial activity against *S. aureus* and *L. monocytogenes*, with MIC values of 0.31 mg/mL and 0.62 mg/mL, respectively, for all concentrations tested. These results suggest that bovine and human hydrolysates exhibit equivalent and potent antibacterial activity. Furthermore, similar antibacterial activity was observed against *E. coli*, with MIC values of 10 mg/mL, as well as against *Micrococcus luteus*, with MIC values of 5 mg/mL, for all samples.

Finally, the results showed that the antibacterial activity of bovine hemoglobin hydrolysates against *Salmonella Newport* was significantly higher than that of human hydrolysates. In fact, for all the concentrations tested, the MIC of bovine hydrolysates was 5 mg/mL, i.e., half the MIC of 10 mg/mL observed for human hydrolysates. The results obtained have been summarized in Table 2.

#### 2.1.2. Antioxidant Activities of Human and Bovine Hemoglobin Hydrolysates

##### β-Carotene Bleaching Inhibition Activity

In this study, the lipid peroxidation inhibition activity of human and bovine hemoglobin hydrolysates (as a control) was measured by evaluating their ability to prevent linoleic acid oxidation in an emulsified model system. Namely, products of lipid oxidation can cause protein oxidation, while carbohydrates are less sensitive to oxidation [24].

The antioxidant activity of human and bovine hemoglobin hydrolysates (as a control) was measured using the β-carotene bleaching assay, shown in Figure 2.

The effects of various doses of hydrolysates were evaluated and contrasted with those of a synthetic bioactive peptide produced from hemoglobin hydrolysis, neokyotorphin (NKT, 0.5 mg/mL), as well as a standard antioxidant, BHT (butylated hydroxytoluene) (0.5 mg/mL). Depending on their concentration, all hydrolysates inhibited β-carotene oxidation to varying degrees. Antioxidant activity nearing 100% signifies comparable efficacy to the reference antioxidant BHT (100 ± 1.12%).

The results showed that all hydrolysates inhibited β-carotene oxidation to different degrees depending on the concentration, with a relative antioxidant activity (RAA) of NKT that is 84.95 ± 5.63%, statistically lower (*p* < 0.05) but close to that of BHT. Thus, this peptide has a lipid protection capacity close to that of BHT.

Furthermore, there was not a significant difference (*p* > 0.05) in the antioxidant activity of human and bovine hemoglobin hydrolysates at any dosage of 1 mg/mL, 5 mg/mL, or 10 mg/mL, independent of the initial substrate percentage (at 1, 2, 8, or 10%). In addition, the relative antioxidant activity (%) increased with increasing protein hydrolysate concentration. Among the tested concentrations, it was observed that a concentration of 10 mg/mL displayed significantly higher antioxidant activity compared to other concentrations. For human samples, the respective antioxidant activities were recorded as 99.26 ± 0.81%, 98.14 ± 6.35%, 98.9 ± 1.23%, and 98.7 ± 9.17% for concentrations of 1%, 2%, 8%, and 10% (*p* < 0.05). Similarly, for bovine samples, the corresponding antioxidant activities were 99.3 ± 3.6%, 100.6 ± 1.1%, 98.8 ± 2.6%, and 99.8 ± 1.5% for concentrations of 1%, 2%, 8%, and 10% (*p* < 0.05). Notably, at a concentration of 10 mg/mL, the observed antiradical activity closely resembled that of BHT at 0.5 mg/mL. These findings suggest that these products have the ability to protect lipids and can be considered primary antioxidant scavengers of free radicals in emulsion. The antioxidant activity does not significantly differ between bovine and human hemoglobin hydrolysates, as shown in the dose-effect curves A and C.

It is interesting to note that, when compared to BHT at 0.5 mg/mL, the relative antioxidant activity of phenolic compounds extracted from olive oil in the traditional manner is 67.40 ± 1.02%, which can be used to compare the study’s findings with those of other substances [25]. The antioxidant activity of our hydrolysates, which were dissolved at a concentration of 5 mg/mL, corresponds approximately to this.

##### DPPH’s Free Radical Scavenging Capacity

The method of DPPH radical scavenging activity is based on the ability of our hydrolysates to capture the DPPH● radical. It is a spectrophotometric method at 517 nm following the absorption decrease recently described by Molyneux [26].

The results presented in Figure 3 clearly demonstrate that both human and bovine hydrolysates have high DPPH free radical scavenging activity. An significative increase in their antioxidant activity is observed as the peptide concentration increases (1, 5, and 10 mg/mL). The results obtained are in agreement with the conclusions reported in the previous work carried out by [23], who observed that DPPH scavenging activity increased with increasing concentrations of 1% (*w*/*v*) bovine hemoglobin hydrolysates (2.5–10 mg/mL) using pepsin as well.

The results presented in the following figures showed no significant difference (*p* > 0.001) between bovine and human hydrolysis samples for concentrations of 1, 5, and 10 mg/mL, regardless of the initial increase in bovine and human hemoglobin concentrations (1 to 10% (*w*/*v*)). When the concentration was fixed at 10 mg/mL, the antioxidant activity was significantly higher (*p* < 0.05) than at other concentrations. The obtained results were 75.8 ± 0.70, 74.9 ± 1.44, 75.9 ± 0.64, and 77.9 ± 4.70 for 1, 2, 8, and 10%, respectively, for human samples, and 76.9 ± 1.67, 73.9 ± 2.33, 74.3 ± 2.08, and 75.5 ± 0.70, respectively, for bovine samples.

Second, the results of the DPPH free radical scavenging activity of the hemoglobin hydrolysates were calculated in terms of the Trolox equivalent antioxidant capacity (TEAC) and half-maximum inhibitory concentrations (IC_50_). The IC_50_ value, which represents the concentration of a sample needed to achieve 50% free radical inhibition, is typically used to assess free radical scavenging activity. The capacity to scavenge free radicals increases as the IC_50_ value decreases. The tested hemoglobin hydrolysates’ IC50 and TEAC values were compared, and the results are shown in Table 3.

The results showed that hydrolysates from human and bovine hemoglobin had similar effects on the ability to trap the DPPH radical (*p* > 0.05), with IC_50_ for 1, 2,8, and 10% not differ significantly. Regarding TEAC, the same conclusion can be drawn, the values in humans did not differ significantly (*p* < 0.05) compared to bovine regardless of the increase in substrate concentration, from 1 to 10% (*w*/*v*). When the results were compared with Trolox, it was observed that the DPPH radical scavenging activity in bovines and humans was lower, with a TEAC value of less than 1. In addition, the activity of NKT was found to be higher, more than twice that of peptide hydrolysates, with a value of 0.45 ± 0.07.

##### Antioxidant Properties Products by ABTS Assay

The ABTS radical scavenging method is based on the molecules’ ability to inhibit cation radicals. This consists of a measurement of antioxidant activity by spectrophotometric tracking of the discoloration of the reaction mixture, as described by Re et al. [27]. Trolox is used as a reference for comparisons. The results presented in Figure 4 show that both human and bovine hydrolysates have a strong ABTS radical inhibition capacity. As the hemoglobin hydrolysate concentration increases (1, 5, and 10 mg/mL), a significant increase in antioxidant activity is observed. This assay also confirms the results obtained in previous work conducted by Abou-Diab et al. [23]. The following figures show no significant difference (*p* > 0.05) between bovine and human hydrolysis samples for the concentrations of 1, 5, and 10 mg/mL, regardless of the increase in the initial concentration of bovine and human hemoglobin substrate (1–10% (*w*/*v*)). The antioxidant activity of the substrate was found to be considerably higher (*p* > 0.05) at 10 mg/mL compared to the other doses. The results of the IC_50_ and TEAC comparative tests of the hydrolysates are presented in Table 4.

It was necessary to use concentrations of human hemoglobin hydrolysates of 3.24 ± 0.38 to 3.56 ± 0.45 mg/mL to inhibit 50% of ABTS radicals, whereas for bovine hemoglobin hydrolysates, it was between 3.74 ± 0.53 and 4.38 ± 0.56 mg/mL. The results of the statistical analysis revealed no significant difference (*p* > 0.05) in IC_50_ values between the two types of hydrolysates, although the values were very close. Regarding NKT, a concentration of 0,64 ± 0.06 mg/mL was required to inhibit 50% of ABTS radicals, which is close to the value of Trolox, but statistically different (*p* < 0.05). Both bovine and human hemoglobin hydrolysates exhibited lower radical scavenging activity compared to Trolox, as indicated by their TEAC values being less than 1. However, the reference bioactive peptide, NKT, demonstrated a very similar radical scavenging activity to Trolox, with a value of 0.90 ± 0.05.

These results indicate that bovine and human hemoglobin hydrolysates (from 1 to 10%) have antioxidant properties, but their efficiency is lower than that of Trolox because of the competitiveness between the many bioactive peptides they contain.

##### Evaluation of Total Antioxidant Capacity

To evaluate the total antioxidant capacity of the hydrolysates, the TEAC unit (mg/mL) was used at three different concentrations (1, 5, and 10 mg/mL), using Trolox as the reference antioxidant. The results obtained from Figure 5 show a direct relationship between the hydrolysate concentration and the increase in antioxidant capacity. Consistent with expectations, it was found that the concentration of 10 mg/mL displayed significantly superior antioxidant activity compared to the other concentrations that were tested. This observation held true for both human and bovine samples.

Furthermore, it was observed that bovine hydrolysates showed a higher TEAC compared to human hydrolysates for most of the concentrations tested (1, 5, and 10 mg/mL) and regardless of the initial substrate concentration, from 1 to 10% (*w*/*v*). For BHT (0.5 mg/mL), an antioxidant activity of 2.02 ± 0.007 mg/mL was found under the same conditions. The TEAC of all hydrolysates, whether of human or bovine origin, at 10 mg/mL is approximately two times lower than that of BHT at 0.5 mg/mL. Therefore, the antioxidant activity of the 10 mg/mL hydrolysate is equivalent to 0.25 mg/mL BHT by the phosphorus-molybdenum method.

### 2.2. Study of the Antimicrobial, Antioxidant Activity and Peptidomic Analysis of Fractions of Peptide Hydrolysates

The results of the study on the antimicrobial and antioxidant activity and the peptidomic analysis of the peptide fractions obtained after 3 h of hydrolysis of human and bovine hemoglobin with a degree of hydrolysis of DH = 10% are presented in the different figures. In Figure 6A, the chromatographic hydrolysis profiles of bovine and human hemoglobin are shown. Fraction one (1) corresponds to the NKT peptide (α137-141, neokyotorphin) of interest, and subsequent fractions were collected every 5 min. The aim was to identify the most active fractions to add value to these high-added-value compounds. Table 5 shows the results of the antimicrobial activity of the peptide fractions against six bacterial strains. The results show that fraction 1 exhibits strong antibacterial activity with a MIC of 1 µg/mL against *Kocuria rhizophila* and *Listeria monocyte* genes, as well as a MIC of 2 µg/mL against *Staphylococcus aureus*, 4 µg/mL against *Salmonella Newport*, and finally 8 µg/mL against *Escherichia coli* and *Micrococcus luteus*. In both human and bovine hemoglobin, fraction 1 (representing NKT) has one of the highest levels of antimicrobial activity.

The last fraction, 9, corresponds to heme, which has not been hydrolyzed and remains unchanged. Numerous studies have reported good bacterial activity in heme [10]. However, it is interesting to note that all fractions showed antibacterial activity, although the strength of the activity varied between strains. For some bacterial strains, fractions 3, 4, 5, and 6 showed lower MICs than fractions 2, 7, or 8. Their inhibitory effect against these bacteria is higher.

Antioxidant activity was assessed using the ABTS^+^ radical scavenging method and is presented in Figure 6B. Fraction 1, containing NKT, again shows very promising antioxidant activity with over 75% inhibition of ABTS^+^. These results are consistent with previous studies [3,24]. Fractions 4, 5, 6, and 7 demonstrate higher antioxidant activity in human and bovine hemoglobin, with more than 50% inhibition of ABTS^+^. In contrast, the other fractions show moderate activity, but with more than 25% inhibition of ABTS^+^.

In studies by [28,29], the ‘Zipper’ mechanism produced a wide variety of peptides from bovine and human hemoglobin hydrolysate. Nevertheless, a comparative analysis of the peptide populations in the fractions was performed by UPLC-MS/MS. In order to present the results clearly, a histogram (Figure 6C) showing the number of unique peptide sequences identified was used. This allows clear and precise visualization of the differences and similarities between the peptide populations in the fractions. Furthermore, by cross-referencing the results, it was observed that the same fractions exhibited both antioxidant and antibacterial properties in both species, notably fractions 4, 5, 6, and 7. This observation can be explained by the greater number of peptide sequences identified in these fractions.

## 3. Discussion

### 3.1. Analysis of the Antimicrobial and Antioxidant Activity of Bovine and Human Hemoglobin Hydrolysates and Their Peptide Fractions

In the first part of our study, we examined the antimicrobial and antioxidant activities of bovine and human hemoglobin hydrolysates obtained by treatment with pepsin. The results showed that all hemoglobin hydrolysates, regardless of origin, exhibited antimicrobial activity against the six bacterial species tested, with a clear zone of bacterial growth inhibition observed at a concentration of 20 mg/mL. Concentration proved to be a key factor in this activity, irrespective of the amount of hemoglobin substrate used (1%, 2%, 8%, or 10% *w*/*v*). The determination of the minimum inhibitory concentration (MIC) validated the antimicrobial activity of human and bovine hemoglobin hydrolysates. The peptides derived from these hydrolysates showed activity against Gram-positive and Gram-negative bacteria. In particular, peptides active against *Kocuria rhizophila, Staphylococcus aureus*, and *Listeria monocytogenes* had MIC values significantly lower than those of *Escherichia coli* and *Salmonella Newport*. The antimicrobial activity of human and bovine hydrolysates was found to be equivalent, except for *Salmonella Newport*, for which bovine hydrolysates proved more potent.

These results highlight the potential of bovine and human hemoglobin hydrolysates as effective antimicrobial agents against microorganisms of interest to the food industry. These microbial strains are frequently implicated in the biological deterioration of foodstuffs, both during production and distribution [23]. These results are in agreement with those obtained previously using the agar diffusion method and are also consistent with previous studies [23], who also observed an inhibitory effect of bovine or porcine hemoglobin hydrolysates on bacteria such as *M. luteus*, *L. Innocua*, *E. coli*, and *S. Newport*. Peptides derived from hemoglobin could be promising candidates for food preservation. Their antimicrobial properties could help extend the shelf life of food products, reducing the risk of bacterial contamination and spoilage [3].

To assess the antioxidant activity of human and bovine hemoglobin hydrolysates, we used different methods. The first method is the bleach inhibition activity of beta-carotene. The hydrolysates showed a high antioxidant activity, comparable to that of the reference antioxidant BHT. Next, the DPPH free radical scavenging activity was assessed, showing that the hydrolysates had high antioxidant activity. Another method used was the assessment of antioxidant activity by the ABTS assay, which confirmed that the hydrolysates had a strong capacity to inhibit ABTS radicals. Finally, the results for total antioxidant capacity also confirmed their potential as antioxidants, although they were less effective than BHT. For all 4 types of tests used in this study, both human and bovine hemoglobin hydrolysates exhibited significant antioxidant properties, indicating their potential as antioxidants regardless of the initial substrate concentration, from 1 to 10% (*w*/*v*). Importantly, there was no significant difference (*p* > 0.05) between bovine and human hydrolysis samples for concentrations of 1.5 and 10 mg/mL, irrespective of the increase in initial concentration of bovine and human hemoglobin substrate (1–10% (*w*/*v*)). The antioxidant activity of the substrate was significantly higher (*p* > 0.05) at 10 mg/mL than at the other doses.

Therefore, hydrolysis is an important step to generate smaller bioactive peptides that possess biological activity, such as significant antiradical activity. It is important to note that the antioxidant properties of peptides are influenced by various factors such as their molecular size, composition, hydrophobicity, and the electron transfer capacity of the amino acid residues in the sequence [28]. Finally, it is interesting to note that these results are in agreement with those obtained previously in studies [23], who also observed their strong potential as antioxidants. These results suggest that human and bovine hemoglobin hydrolysates can be used as antioxidant agents to combat the free radicals that cause cellular damage and disease.

The second phase of our study focused on demonstrating the antimicrobial and antioxidant properties of the human hemoglobin and bovine hemoglobin hydrolysate fractions. Figure 6A shows the chromatographic profiles of the hydrolysis of the two types of hemoglobin, with fraction 1 containing the NKT peptide (α137-141, neokyotorphin) of particular interest. Subsequent fractions were collected at 5 min intervals to identify those with the highest activity to exploit these high-value compounds.

The antimicrobial activity results show that fraction 1 has strong antibacterial activity, with MIC (Minimum Inhibitory Concentration) values of 1 to 8 µg/mL depending on the bacterial strains. This fraction has one of the highest levels of antimicrobial activity in both human and bovine hemoglobin.

The results of this study confirm what has been reported in previous studies [21,22]. They also observed a potent inhibitory effect of the α137-141 peptide (NKT) against Gram-positive bacteria such as *Staphylococcus aureus*, *Listeria innocua,* and *Micrococcus luteus*, as well as Gram-negative bacteria such as *Escherichia coli* and *Salmonella enteritidis*. Compared to other well-known antimicrobial peptides derived from bovine hemoglobin, such as the α1-23 and α1-32 fragments, α137-141 showed up to 54-fold lower minimum inhibitory concentrations (MICs) for a given strain, indicating considerably more potent antimicrobial activity, despite its shorter amino acid sequence [21,22].

Furthermore, the peptide α137-141 has been identified as a natural preservative for meat during storage or distribution [3]. Tests showed that it reduced rancidity time by 60% and inhibited microbial growth after 14 days of refrigeration. The study also revealed that its effectiveness is comparable to that of butylated hydroxytoluene (BHT), a synthetic preservative commonly used in the food industry. This discovery opens up promising prospects for the food industry by offering a natural alternative to chemical preservatives.

Our study also demonstrated the successful identification and isolation of the α137-141 peptide from human hemoglobin, without modification [28], and confirmed its highly antimicrobial potency, equivalent to its counterpart isolated from bovine hemoglobin. These results reinforce the potential of the α137-141 peptide as a promising candidate for the development of natural preservatives and antimicrobial alternatives in various food and pharmaceutical applications.

Finally, all the peptide fractions demonstrated antibacterial activity, although their intensity varied depending on the bacterial strain. Some fractions showed higher inhibitory activity than others. As for antioxidant activity, fraction 1 containing NKT showed very promising activity, followed by fractions 4, 5, 6, and 7, with more than 50% inhibition of the ABTS radical. The other fractions also showed moderate antioxidant activity, with over 25% inhibition of this radical.

### 3.2. Identification of Antimicrobial Peptides Resulting from the Hydrolysis of Human and Bovine Hemoglobin

By performing enzymatic hydrolysis using pepsin on bovine and human hemoglobin, the results obtained by UPLC-MS/MS revealed that some of the peptides identified display antimicrobial and antioxidant properties, as reported in previous studies [15,23,29,30,31], and are listed in the Table 6. These results enabled a comparison of peptides between the two species and highlighted similarities or differences in their composition. Most of the peptides identified in cattle were found in humans without modification, but some antimicrobial peptides present in bovine hemoglobin were not found in their original form in human hemoglobin. This difference could be due to differences in the cleavage sites or to modifications in the peptide sequence of the two types of hemoglobin. For example, the antimicrobial peptide KLLSHSL located at α99-105 became KLLSHCL in human hemoglobin, as did the peptide KLLSHSLL, which became KLLSHCLL located at α99-106.

The results of this study corroborate the conclusions reported by Outman et al. [28] in the scientific literature. These authors carried out in silico analyses that revealed great similarities between bovine and human hemoglobin. Sequence comparison analyses showed a high identity of 88% and 85% for the α and β chains, respectively, underlining a strong similarity between the two species. In addition, these analyses showed that during enzymatic hydrolysis, there was major conservation of the number and location of cleavage sites between the two types of hemoglobin, although there were also differences identified. Interestingly, both types of hemoglobin were subjected to similar enzymatic hydrolysis conditions (23 °C, pH 3.5), and the reaction mechanisms were comparable, both following a “zipper” mechanism.

Following these observations, the results revealed the presence of several new bioactive peptides in human hemoglobin, some of which were already known, while others were discovered for the first time. These include antibacterial peptides such as α37-46 PTTKTYFPHF, α36-45 FPTTKTYFPH, α137-141 TSKYR, and α133-141 STVLTSKYR, as well as an antioxidant peptide α137-141 TSKYR [28].

It is also interesting to note that the identified peptides are present in the same bovine and human protein fractions, suggesting that their release occurred simultaneously during the hydrolysis process. However, it is difficult to determine which fraction is more active in terms of bioactive peptides, for two main reasons. Firstly, although there are more than 200 peptides identified for each species at this stage of hydrolysis, only a few have been reported in the literature to have antibacterial properties. On the other hand, the degree of hydrolysis (DH) influences the composition of the peptide populations obtained, as the molecular weight of peptides decreases as DH increases [3]. Whereas in this study, hydrolysis was only performed at a DH of 10 and was not performed at other DHs.

These findings open up new prospects for research into the properties and potential applications of these peptides in various fields, particularly in the food and pharmaceutical industries. These new bioactive peptides identified in human hemoglobin could be exploited for their beneficial properties as antibacterial and antioxidant agents, offering new opportunities for the development of innovative and natural products.

## 4. Materials and Methods

### 4.1. Materials: Chemicals and Cultures Used

The chemicals used for the dosages are: 1,1-diphenyl-2-picrylhydrazyl (DPPH), butylated hydroxytoluene (BHT), 2,2′-azino-bis 3-ethylbenzothiazoline-6-sulfonic acid (ABTS), 6-hydroxy-2,5,7,8-tetramethylchroman-2-carboxylic acid (Trolox), β-carotene, and linoleic acid were procured from Sigma-Aldrich (Saint-Quentin Fallavier, France). A Milli-Q system was employed in the lab to produce ultra-pure water.

Purified bovine hemoglobin powder (H2625), dark brown, and purified human hemoglobin (H7379), dark red, were obtained from Sigma-Aldrich. Hemoglobins were stored at 4 °C before use. Standard neokyotorphin (α137-141) was purchased from GeneCust (Boynes, France) and stored at −20 °C before use. Pepsin, a lyophilized powder from porcine gastric mucosa, was purchased from Sigma-Aldrich (P6887, Saint-Quentin Fallavier, France). Pepsin activity was measured at 3250 AU/mg protein according to a protocol established by the supplier Sigma-Aldrich (Saint-Quentin-Fallavier, France). Storage at −20 °C.

Bacterial species used: *Listeria monocytogenes* (ATCC 19112), *Staphylococcus aureus* (ATCC 13709), *Micrococcus luteus* (ATCC 9341), *Escherichia coli* (ATCC 8733), and *Salmonella Newport* (ATCC 6962) were from the American Type Culture Collection (ATCC, Rockville, MD, USA). *Kocuriarhizophila* (CIP 53.45)was from the Collection de l’Institut Pasteur (CIP, Paris, France).

### 4.2. Preparation of Bovine Hemoglobin Hydrolysates

Conventional enzymatic hydrolysis using pepsin was used to compare two types of hemoglobin: bovine and human hemoglobin. Bovine hemoglobin is considered a “control” because it has been shown in several enzymatic hydrolysis studies to produce active peptides, including the peptide α137-141, which has various biological activities [16,21,23,31]. The results obtained were compared to this control.

#### 4.2.1. Preparation of the Stock Solution

The stock solution was made by dissolving 15 g of purified human hemoglobin HH (or bovine hemoglobin BH) in 100 mL of ultrapure water. The solution was centrifuged at 6000× *g* for 30 min to recover the supernatant (Eppendorf AG, 22,331 Hamburg, Centrifuge 5804 R, Brinkmann Instruments, Westbury, NY, USA). The effective concentration of HH (CHH) or BH (CBH) was calculated using the Drabkin method. A purified hemoglobin solution with precise concentrations of 1, 2, 8, and 10% (*w*/*v*) was prepared from the measured concentration.

#### 4.2.2. Hydrolysis Process

In order to denature the hemoglobin solution, which was initially in its native, so-called “globular” form, the pH was adjusted to 3.5 using 2 M hydrochloric acid, which was added gradually. Pepsin (EC 3.4.23.1, 3200–4500 units’ mg^−1^ of protein), previously solubilized in ultrapure water with an enzyme/substrate ratio of 1/11 (mole/mole), was added to begin the hydrolysis reaction. The samples were taken at T 0, T 2.5, T 30, T 60, T 120, and T 180 min of hydrolysis, corresponding to different degrees of hydrolysis.

Then, the peptic hydrolysis reaction is stopped by adding 5 M NaCl sodium chloride to a final pH of 9, which deactivates the enzyme. Throughout the reaction, the temperature was maintained at a constant 30 °C.

After enzymatic hydrolysis of bovine and human hemoglobin by pepsin, the samples (triplicates) were subjected to lyophilization, yielding a powder that was used for the antibacterial and antioxidant tests. For these tests, different concentrations (1, 5, and 10 mg/mL) were obtained by diluting the powders of bovine and human hemoglobin hydrolysates.

#### 4.2.3. Fractionation of Peptide Hydrolysates by Semi-Preparative HPLC

In order to refine the research results, fractions were performed. To obtain more active peptides in one step, it was necessary to increase the concentration of the initial substrate. For this purpose, a hydrolysate could be selected with a degree of hydrolysis of 10% and hydrolysis times of 3 h. It was shown that increasing the concentration of peptides in the 10% (*w*/*v*) hydrolysate allowed the recovery of up to 10 times more active peptides, such as the α137-141 peptide, compared to the initial hydrolysate. This suggests an interesting valorization of the co-products, with a 10-fold enrichment [3]. Fractions were collected every 5 min using a liquid chromatography system. Chromatographic data were traced, acquired, and analyzed using Waters software. Chromatographic procedures were performed with a semi-preparative C4 column (RP 2.6 UM 150 × 2.1 MM). The mobile phases were ultrapure water/trifluoroacetic acid (99:1, *v*/*v*) as solvent A and acetonitrile/trifluoroacetic acid (99:1, *v*/*v*) as solvent B. Samples were filtered at 0.20 μm and then injected. Online UV absorbance scans were performed between 200 and 390 nm at a rate of one spectrum per second with a resolution of 1.2 nm [17,31]. The injection volume was 60 µL, and the flow rate was 0.6 mL/min. A gradient was applied with solvent B increasing from 5% to 30% in 30 min, then to 60% for 10 min, and maintained until 47 min at 95%, then back to initial conditions. The tubes containing the different fractions were dried with SpeedVac and stored at −20 °C.

#### 4.2.4. Analysis RP-UPLC, Mass Spectrometry Analysis

Aliquots of 10 µL of human and bovine hydrolysates, each with a concentration of 30 mg/mL, were centrifuged for 10 min. at 8000× *g* before being subjected to RP-HPLC-MS/MS analysis in triplicate. A C18 column (150 mm × 3.0 mm, 2.6 m, Uptisphere CS EVOLUTION, Interchim, France) was used to separate the peptides. Solvent A (0.1 *v*/*v* formic acid/99.9 *v*/*v* water) and Solvent B (0.1 *v*/*v* formic acid/99.9 *v*/*v* acetonitrile (ACN)) made up the mobile phases. The ACN gradient (flow rate 0.5 mL·min^−1^) was as follows: from 5% to 30% solvent B over 40 min, from 30% to 100% solvent B over 10 min, followed by washing and equilibrating procedures using successively 100% and 1% solvent B for 5 min each.

The eluate was directed into the electrospray ionization source of the qTOFSynapt G2-Si™ (Waters Corporation, Manchester, UK), previously calibrated using a sodium format solution. Mass spectrometry (MS) measures were performed in sensitivity, positive ion, and data-dependent analysis (DDA) modes using the proprietary Mass Lynx software (Waters).

The capillary and cone voltages were set to 3000 and 60 V, respectively, while the source temperature was set at 150 °C. With a scan time of 0.2 s, MS data were gathered for *m*/*z* values between 50 and 2000 Da. For the MS/MS study, a maximum of 10 precursor ions with a 10,000-intensity threshold were used. When collecting MS/MS data, collision-induced dissociation (CID) fragmentation mode was used. A scan period of 0.1 s was specified, and the stated voltage ranges for the lower and higher molecular mass ions, respectively, were 8 to 9 V and 40 to 90 V. Every two minutes for 0.5 s, leucin + enkephalin ([M + H]^+^ of 556.632]) were injected into the system to monitor and rectify the measurement error throughout the analysis.

The chromatographed peptides were identified using protein database searches in the UniProt databases that were limited to *Bos taurus* and *Homo sapiens* (access online on July 2022) using PEAKS^®^ Studio XPro (v 10.6) (Bio Informatics Solutions Inc., Waterloo, Canada). A mass tolerance of 35 ppm and an MS/MS tolerance of 0.2 Da were permitted. With pepsin designated as the hydrolysis enzyme, data searches were carried out. The relevance of protein and peptide identities was judged according to their identification score in the research software (using a *p*-value < 0.05 and a false discovery rate <1%).

### 4.3. Determination of the Hydrolysates Bioactivity

#### 4.3.1. Antibacterial Activity

##### Antimicrobial Test on Petri Dish: Agar Diffusion Method

The determination of the antimicrobial activity of hydrolysates of human and bovine hemoglobin was realized according to the method of Adje et al. [32]. This is a method based on the diffusion of the antibacterial agent within an agar medium seeded with a target strain. A total of six bacterial strains were tested for antibacterial activity: *Staphylococcus aureus* (ATCC 13709), *Listeria monocytogenes* (ATCC 19112), *Micrococcus luteus* (ATCC 9341), *Kocuriarhizophila* (CIP 53.45), *Escherichia coli* (ATCC 8733), and *Salmonella Newport* (ATCC 6962). It is important to note that these bacterial strains are commonly responsible for food spoilage [21,33]. These species are stored at −20 °C in a nutritive broth containing glycerol. The pre-culture of the species was performed by inoculation: 50 µL of bacteria were added to 5 mL of Müeller–Hinton medium (MH), and the whole was incubated at 37 °C under agitation (60 rpm) for 24 h. Then, the absorbance of the cultures was determined with a spectrophotometer at 620 nm using the MH medium alone as a blank. In order to obtain an absorbance of 0.25 at 620 nm, a series of dilutions were performed in Tryptone Salt (TS), according to the type of bacteria, to obtain a final Colony Forming Unit CFU/mL. (CFU) to ×10^6^: 1/1000 dilution for gram (-) bacteria and 1/100 dilution for gram (+) bacteria. The MH agar medium, previously casted and dried in Petri dishes, was inoculated by the inundation method: one milliliter of complete inocula from the last dilutions (1/100 or 1/1000) was put in contact with the agar for 20 min. The excess inoculum was removed by aspiration with a sterile pasteur pipette. These dishes were then dried under a laminar flow host for more than 30 min. Then, a spot deposit of 10 µL of 0.25 μm filtered samples, as well as the positive control, was deposited on the agar surface. After drying, the plates were incubated for 24 h at 37 °C. The diameter of the clear zone of growth inhibition against a positive control, ampicillin (0.1 mg/mL) for all bacterial species and colistin (0.1 mg/mL) for *Escherichia coli*, was used to quantify the antibacterial activity. This antimicrobial test was run three times.

##### Determination of the Minimal Inhibitory Concentration (MIC)

The MIC (“Minimal Inhibitory Concentration”) is the lowest concentration of the studied peptide hydrolysates that can inhibit the growth of the studied bacterial species.

This is a growth inhibition test in a liquid medium carried out on a sterile 96-well microplate, whose response is read after 24 h of incubation [34]. This test quantifies the antimicrobial activity of a hydrolysate, an isolated peptide, or any other antimicrobial agent. The Mueller–Hinton culture medium was chosen for this assay. Initially, 100 μL of this medium was added to each well and supplemented with 100 μL of peptide hydrolysates (40 mg/mL), leaving a negative control range. A total of 100 μL bacterial suspension pre-diluted in Mueller–Hinton was added last to each well for a final bacterial load of 10^5^ CFU/mL. The negative control corresponds to the medium without bacteria, and the positive control corresponds to 50 µL of culture medium contacted with 50 µL of bacterial suspension.

After incubation at 37 °C for 24 h, the inhibition of bacterial growth was assessed by measuring the absorbance at 600 nm using a microplate reader (Safas, model MP96 UV-Vis Spectrophotometer, Agilent Technologies, Santa Clara, CA, USA). The measurement of MIC values was performed in triplicate. The absorbance values obtained from wells containing peptide hydrolysates were compared to those of negative controls and control culture. All data are presented as the mean ± standard deviation (SD) and represent the average of three replicates.

#### 4.3.2. Antioxidant Activity

In this study, four in vitro chemical tests based on various antioxidant mechanisms were used to determine the antioxidant activity of bovine and human hemoglobin hydrolysates, as well as peptidic fractions and the synthetic peptide neokyotorphin.

##### Antioxidant Assay Using the β-Carotene Bleaching Method

This method consists of spectrophotometrically monitoring the bleaching of β-carotene: the loss of its yellow color resulting from its reaction with radicals formed by the oxidation of linoleic acid in emulsion [35]. The bleaching of β-carotene, measured by the decrease in initial absorbance at 470 nm, is slowed down in the presence of antioxidants, resulting in lower oxidation kinetics. Koleva et al. [36] indicated that the hydrolysate’s capacity to prevent β-carotene bleaching was assessed. A stock solution of β-carotene (4 mg), linoleic acid (100 L), and Tween-40 (800 L) was dissolved in 4 mL of chloroform. After the latter had entirely evaporated at 45 °C under a vacuum in a rotary evaporator (Heidolph, Schwabach, Germany), 400 mL of distilled water was added.

The capability of the hydrolysate to prevent the bleaching of -carotene was assessed, as Koleva et al. [36] mentioned. A stock solution of β-carotene (4 mg), linoleic acid (100 L), and Tween-40 (800 L) was dissolved in 4 mL of chloroform. After the latter had entirely evaporated at 45 °C under a vacuum in a rotary evaporator (Heidolph, Schwabach, Germany), 400 mL of distilled water was added. The mixture was then agitated vigorously. Before each experiment, the β-carotene/linoleic acid emulsion was freshly prepared.

In tubes with a sample (5 mL) of the emulsion were 500 L of each sample that had been prepared using ultrapure water at various concentrations. Each sample’s absorbance at 470 nm was assessed before and after being incubated for two hours at 50 °C (Shimadzu UV-1650 PC Spectrophotometer, Shimadzu Corporation, Kyoto, Japan). Butylated hydroxytoluene (BHT) was used as a positive control at a concentration of 0.5 mg/mL.

Equation (1) was used to calculate relative antioxidant activity:(1)$$RAA(\%)=(\frac{A\;sample}{A\;control})\times 100$$

A sample is the absorbance of samples (with the emulsion), and A control is the absorbance of BHT (with the emulsion). All the data are presented as mean ± SD and are the mean of three replicates.

##### DPPH Radical Scavenging Capacity

According to the recently described method of Bersuder et al. [37], the DPPH (1,1-diphenyl-2-picrylhydrazyl) radical scavenging capacity of the samples was evaluated. An ethanolic stock solution of DPPH (0.02 mass %) as a source of free radicals was prepared 12–16 h before use.

Each sample was diluted in ultrapure water to different concentrations, and a volume of 500 μL was added to 250 μL of 99% ethanol and 375 μL of DPPH solution. The mixtures were vortexed before being incubated for 60 min at 30 °C in the dark. The absorbance of the mixture was measured at 517 nm against a blank consisting of pure ethanol. The scavenging capacity was measured spectrophotometrically (PC Shimadzu UV-1650 spectrophotometer, Shimadzu Corporation, Kyoto, Japan) by following the decrease in absorbance at 517 nm. The reaction mixture’s lower absorbance indicates higher DPPH free radical scavenging activity. The same operational conditions were employed to compare BHT. The radical scavenging activity (RSA) is quantified as a percentage and calculated using the following Formula (2).
(2)$$DPPH\;radical\;scavenging\;activity\left(\%\right)=\frac{\left(A\;blank-A\;sample\right)}{A\;control}\times 100$$
where A blank is the absorbance of the control reaction (containing all reagents except the sample), A sample is the absorbance of samples (with the DPPH solution), and A control is the absorbance of BHT (with the DPPH solution).

The DPPH+ radical scavenging activity was expressed using the Trolox Equivalent Antioxidant Capacity (TEAC) values and half-maximal inhibitory concentrations (IC_50_). The IC_50_ was graphically estimated using linear regressions of the percentage curves of inhibition in response to various concentrations of the hydrolysates under test. It shows the sample volume needed to inhibit 50% of DPPH radicals and the value decreased as the compound’s anti-free radical activity increased [38]. The concentration of standard Trolox, a water-soluble vitamin E analog, was utilized to calculate the TEAC value [39]. Standard Trolox had the same antioxidant capacity as a 1 mg/mL solution concentration of the antioxidant molecule under research.

In biology, it serves as a reference antioxidant for evaluating oxidative stress resistance. The results of each analysis, which were performed in triplicate, were provided as mean values with standard deviations.

##### Antioxidant Properties Products by ABTS Assay

The method outlined by Re et al. [27] was used to test the radical scavenging abilities of ABTS (2,2′-azino-bis 3-ethylbenzothiazoline-6-sulfonic acid). Aqueous interactions between 7 mM ABTS and 4.95 mM potassium persulfate formed the cationic radical ABTS^+^. This was prepared 12 to 16 h before use. The solution was stable when kept at ambient temperature and protected from light. A portion of this solution was diluted with ethanol at 30 °C in order to attain an absorbance at 734 nm of less than one. The hydrolysates were first dissolved in various concentrations of ultrapure water, and then 10 μL of each sample was combined with 1 mL of the diluted ABTS^+^ solution. Exactly 6 min after the initial mixing, the absorbance of ABTS^+^ was determined at 734 nm and 30 °C using a spectrophotometer (Shimadzu UV-1650 PC Spectrophotometer, Shimadzu Corporation, Kyoto, Japan). In each test, the proper solvent blanks were used. According to Equation (3), the result is expressed as the percentage inhibition:(3)$$ABTS+radical\;scavenging\;activity\left(\%\right)=1-(\frac{A\;sample}{A\;blank})\times 100$$

The absorbance of samples (using the ABTS solution) is represented by A sample, while the absorbance of the control reaction (containing all reagents except the sample) is represented by A blank.

The ABTS+ radical scavenging activity was expressed using the half-maximal inhibition concentration (IC_50_) and TEAC values. The data presented includes the mean and standard deviation (SD) of three replicates.

##### Evaluation of Total Antioxidant Capacity

The hydrolysates’ total antioxidant capacity was assessed using the technique described by Prieto et al. [40]. Based on an oxidation-reduction reaction, it is a simple and effective approach for evaluating antioxidant activity.

A phosphate/molybdenum (Mo3O16P4) green complex is formed at an acidic pH as a result of the reduction of molybdenum Mo, which is present as molybdate MoO4^−2^ ions, to molybdenum MoO^2+^ in the presence of the hydrolysate. With the help of this assay, we were able to determine the peptides’ antioxidant status by using hemoglobin hydrolysates as reducing agents in a colorimetric redox reaction. Prior to mixing with the reagent (0.6 M sulfuric acid, 28 mM sodium phosphate, and 4 mM ammonium molybdate), hemoglobin hydrolysates were first dissolved in ultrapure water at different concentrations (2.5 mg/mL, 5 mg/mL, 10 mg/mL, and 20 mg/mL). An aliquot of 300 μL of each sample was then added. The tubes were covered, and they were incubated in a thermal block for 90 min at 95 °C.

After cooling at room temperature, each sample’s aqueous solution’s absorbance was measured at 695 nm against a control (Shimadzu UV-1650 PC Spectrophotometer, Shimadzu Corporation, Kyoto, Japan). A typical blank solution was made up of 3 mL of reagent solution and the necessary amount of the sample’s solvent, and it was incubated in the same manner as the other samples. The positive standard used was BHT (0.5 mg/mL). For calibration, a Trolox solution with concentrations varying from 0 to 0.75 mg/mL was utilized. Equation (4) from the calibration line is used to represent antioxidant activity as Trolox Equivalent Antioxidant Capacity (TEAC) values (mg/mL):A = a × [C_Trolox_] + b(4)

The absorbance at 695 nm is denoted as A, and the equivalent antioxidant concentration is represented by C (mg/mL). The concentration is determined using the standard range curve equation of a reference antioxidant, such as Trolox. The values of a and b in the equation correspond to the origin and slope of the Trolox calibration line, respectively. All the data are expressed as the mean ± SD and represent the average of three replicates.

### 4.4. Statistical Analyses

For each analysis, triplicates of each condition were performed in three different replications. The data were subjected to a one- or two-way analysis of variance (ANOVA). Tukey tests were run on the data using GraphPad Prism 8.0.2 to determine whether the treatment was statistically different from the others at a probability level of 0.05.

## 5. Conclusions

This study provided convincing evidence for the efficacy of bovine and human hemoglobin peptide hydrolysates in inhibiting the growth of a variety of bacteria, including *S. aureus*, *L. monocytogenes*, *M. luteus*, *E. coli,* and *S. Newport*, as well as *K. rhizophila*, independently of their initial hemoglobin concentrations ranging from 1 to 10% (*w*/*v*). In addition, these hydrolysates also exhibited significant antioxidant properties. A particularly remarkable finding concerned human hemoglobin hydrolysates, which revealed, for the first time, their ability to inhibit the growth of certain bacteria while scavenging free radicals and preventing oxidative damage.

These observations suggest that certain hydrolysate fractions contain powerful peptides that act as effective free radical scavengers and also act as natural antimicrobial agents. These characteristics make them particularly interesting for potential applications in food, cosmetics, and pharmaceuticals, where natural antimicrobial and antioxidant agents are increasingly sought after.

Ultimately, further research could be undertaken to isolate and specifically characterize these active peptides in order to deepen our understanding of their mechanisms of action and develop even more effective bioactive products. By combining the benefits of antimicrobial and antioxidant properties, these hydrolysates offer promising potential for future applications aimed at improving consumer health and well-being.

## Figures and Tables

**Figure 1 ijms-24-13055-f001:**
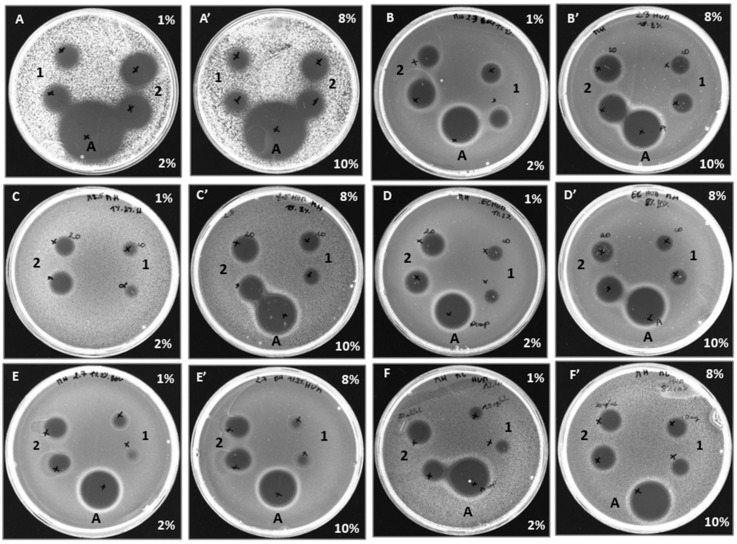
Antibacterial activity of human hemoglobin hydrolysates (1, 2, 8, and 10% (p/v)) obtained with treatment by pepsin. A method based on the diffusion of the antibacterial agent in an agar medium inoculated with a target strain: *Kocuria rhizophila* (**A**,**A’**), *Staphylococcus aureus* (**B**,**B’**), *Listeria monocytogenes* (**C**,**C’**), *Escherichia coli* (**D**,**D’**), *Salmonella Newport* (**E**,**E’**), *Micrococcus luteus* (**F**,**F’**), and 1–2 refer to (10 and 20 mg/mL human HB hydrolysate), respectively, control +: ampicillin (**A**).

**Figure 2 ijms-24-13055-f002:**
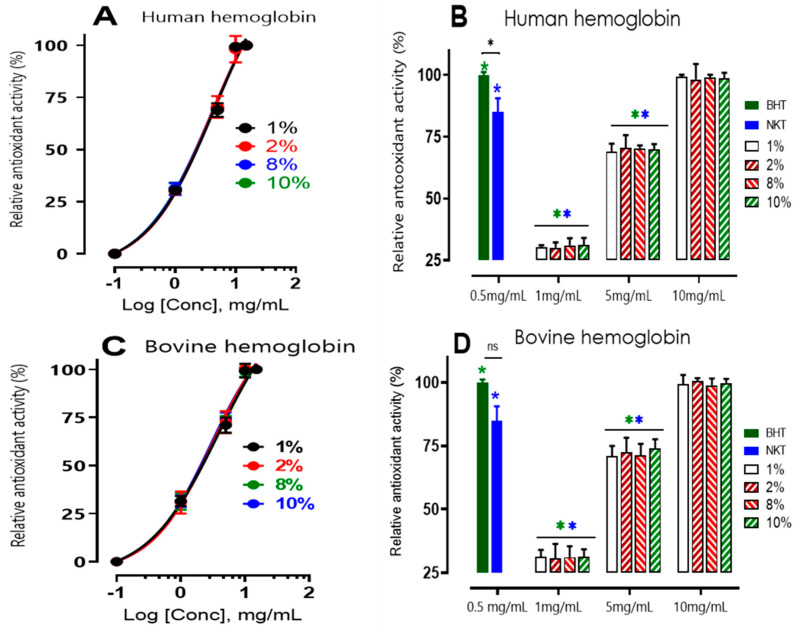
β-carotene bleaching inhibition activity of hemoglobin in human and bovine hydrolysates at different concentrations. (**A**,**C**) show that the variation in relative antioxidant activity (RAA) is dependent on concentration (mg/mL) and not on the initial hemoglobin substrate concentration (1, 2, 8, and 10% (*w*/*v*)). In (**B**,**D**), values marked with a symbol in green (*****) are significantly different from BHT; values marked with a symbol in blue (*****) are significantly different from neokyotorphin (NKT) at the same concentration and under the same conditions; * *p* < 0.05, multiple comparison *t* tests using the Bonferroni–Dum method).

**Figure 3 ijms-24-13055-f003:**
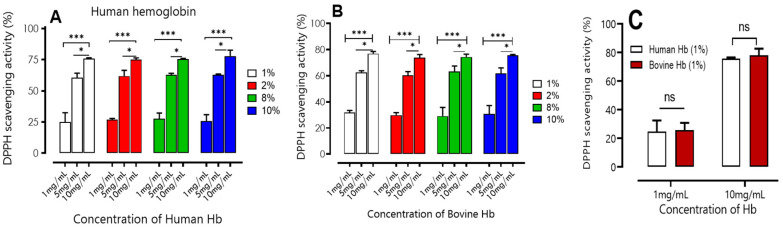
DPPH scavenging activity of human (**A**) and bovine (**B**) hemoglobin hydrolysates at different concentrations (1, 5, and 10 mg/mL) with different initial concentrations of hemoglobin substrate (1, 2, 8, and 10% (*w*/*v*)). Comparisons were made between the different concentrations for each percentage under the same conditions. (**C**) Comparison of DPPH activity of human and bovine hemoglobin hydrolysates at different concentrations (1, 10 mg/mL) at 1%. * *p* < 0.05, *** *p* < 0.001 (one-way ANOVA and Tukey’s multiple comparisons test). ns *p* > 0.05 (paired *t*-test).

**Figure 4 ijms-24-13055-f004:**
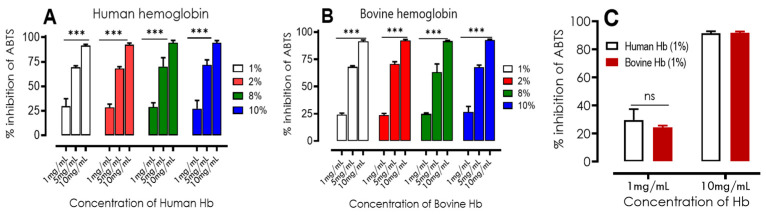
Percentage of ABTS radical inhibition of human (**A**) and bovine (**B**) hemoglobin hydrolysates at different concentrations (1, 5, 10 mg/mL) with different percentages of initial bovine/human hemoglobin substrate of the hydrolysates (1–10%). Comparisons were made between the different concentrations of each percentage under the same conditions. (**C**) Comparison between ABTS radical inhibition of human and bovine hemoglobin hydrolysates at different concentrations (1, 10 mg/mL) at 1% purification of hemoglobin hydrolysates. *** *p* < 0.001 (one-way ANOVA and Tukey’s multiple comparisons test). ns *p* > 0.05 (paired *t*-test).

**Figure 5 ijms-24-13055-f005:**
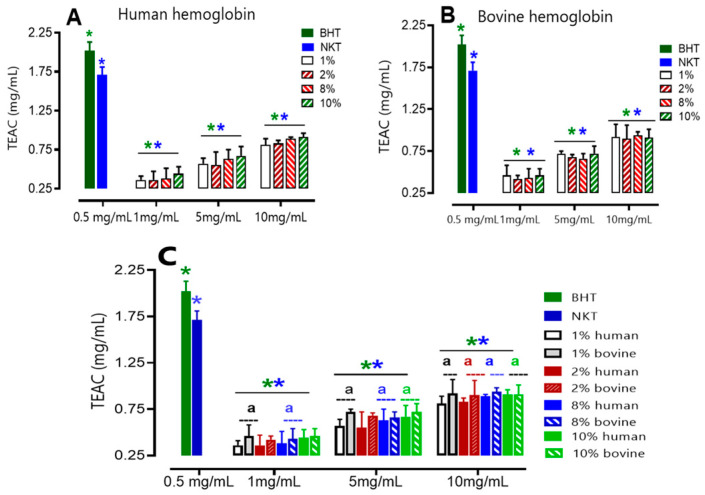
Total antioxidant capacity of human (**A**) and bovine (**B**) hemoglobin hydrolysates at different concentrations. Comparison between humans and bovines (**C**). Values with the symbol (*****) in green are significantly different from BHT; values with the symbol (*****) in blue are significantly different from NKT. * *p* < 0.05, one-way ANOVA followed by Dunnett’s multiple comparison test. Values with a letter (a) in the same concentration are significantly different (*p* < 0.05, Tukey).

**Figure 6 ijms-24-13055-f006:**
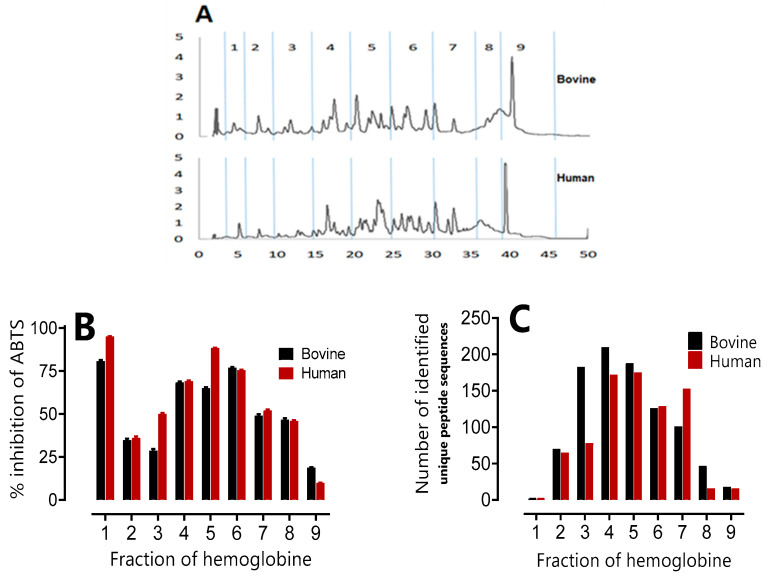
Study of the antimicrobial and antioxidant activity and peptidomic analysis of human and bovine hemoglobin hydrolysate fractions after a 3-h hydrolysis period (pH 3,5, 23 °C, E/S = 1/11, C_BH_ = 10%, C_HH_ = 10%, *w*/*v*). Fractions were collected every 5 min. (**A**) Chromatographic hydrolysis profiles of bovine and human hemoglobin acquired using Empower 3 software (Version 3 Waters) at 215 nm by semi-preparative HPLC, analyzed by a semi-preparative C4 column, (1–9 represent the number of fractions collected). (**B**) Percentage ABTS inhibition of human and bovine hemoglobin hydrolysate fractions at a concentration of 20 mg/mL. (**C**) Peptidomic analysis using UPLC-MS/MS and bioinformatics. Histogram showing the number of unique peptide sequences identified.

**Table 1 ijms-24-13055-t001:** Antibacterial activity of human and bovine hemoglobin hydrolysates obtained with treatment by pepsin and the α137-141 standard.

	Hemoglobin Hydrolysates								
	1%		2%		8%		10%		α137-141
Bacteria Strains	Bovine	Human	Bovine	Human	Bovine	Human	Bovine	Human	Standard
*Kocuriarhizophila*	+++	+++	+++	+++	+++	+++	+++	+++	+++
*Staphylococcus aureus*	+++	+++	+++	+++	+++	++	+++	++	+++
*Listeria monocytogenes*	++	++	++	++	++	++	++	++	++
*Escherichia coli*	++	+	+	+	+	+	++	++	+
*Salmonelle Newport*	++	+	++	+	++	+	++	+	+
*Micrococcus luteus*	++	++	++	++	++	++	++	+	++

Inhibition zones: +++: >1.5 cm; ++: 0.5–1.5 cm, +: <0.5 cm.

**Table 2 ijms-24-13055-t002:** Antibacterial activity of human and bovine hemoglobin hydrolysates.

	Bovine Hemoglobin Hydrolysates (mg/mL)			
Bacteria Strains	1%	2%	8%	10%
*Kocuriarhizophila*	0.15 ± 0.0 ^a^	0.15 ± 0.0 ^a^	0.15 ± 0.0 ^a^	0.15 ± 0.0 ^a^
*Staphylococcus aureus*	0.31 ± 0.0 ^a^	0.31 ± 0.0 ^a^	0.31 ± 0.0 ^a^	0.31 ± 0.0 ^a^
*Listeria monocytogenes*	0.62 ± 0.0 ^a^	0.62 ± 0.0 ^a^	0.62 ± 0.0 ^a^	0.62 ± 0.0 ^a^
*Escherichia coli*	10 ± 0.0 ^a^	10 ± 0.0 ^a^	10 ± 0.0 ^a^	10 ± 0.0 ^a^
*Salmonelle Newport*	5 ± 0.0 ^b^	5 ± 0.0 ^b^	5 ± 0.0 ^b^	5 ± 0.0 ^b^
*Micrococcus luteus*	5 ± 0.0 ^a^	5 ± 0.0 ^a^	5 ± 0.0 ^a^	5 ± 0.0 ^a^
	**Human Hemoglobin Hydrolysates (mg/mL)**			
**Bacteria Strains**	**1%**	**2%**	**8%**	**10%**
*Kocuriarhizophila*	0.15 ± 0.0 ^a^	0.15 ± 0.0 ^a^	0.15 ± 0.0 ^a^	0.15 ± 0.0 ^a^
*Staphylococcus aureus*	0.31 ± 0.0 ^a^	0.31 ± 0.0 ^a^	0.31 ± 0.0 ^a^	0.31 ± 0.0 ^a^
*Listeria monocytogenes*	0.62 ± 0.0 ^a^	0.62 ± 0.0 ^a^	0.62 ± 0.0 ^a^	0.62 ± 0.0 ^a^
*Escherichia coli*	10 ± 0.0 ^a^	10 ± 0.0 ^a^	10 ± 0.0 ^a^	10 ± 0.0 ^a^
*Salmonelle Newport*	10 ± 0.0 ^a^	10 ± 0.0 ^a^	10 ± 0.0 ^a^	10 ± 0.0 ^a^
*Micrococcus luteus*	5 ± 0.0 ^a^	5 ± 0.0 ^a^	5 ± 0.0 ^a^	5 ± 0.0 ^a^

After 24 h of incubation at 37 °C, the minimum inhibitory concentration (MIC) of bovine and human hemoglobin hydrolysates was determined using a microtiter plate assay method. The MICs of bovine and human hemoglobin hydrolysates are compared for each bacterial strain. ^a,b^: The population means for each bacterium in each column with different letters are significantly different, a similar letter means no significant difference, *p* < 0.05 (ANOVA, Tukey).

**Table 3 ijms-24-13055-t003:** Trolox equivalent antioxidant capacity (TEAC).

Purification		1%	2%	8%	10%	NKT	TROLOX
IC_50_ (mg/mL)	Human	3.07 ± 0.34 ^ns^	2.63 ± 0.31 *	2.45 ± 0.28 *	2.68 ± 0.32 ^ns^	0.83 ± 0.09	0.38 ± 0.03
	Bovine	3.83 ± 0.40 ^ns^	3.26 ± 0.36 *	3.01 ± 0.32 *	3.22 ± 0.38 ^ns^		
TEAC	Human	0.12 ± 0.05 ^ns^	0.14 ± 0.04 ^ns^	0.15 ± 0.05 ^ns^	0.14 ± 0.04 ^ns^	0.45 ± 0.07	1
	Bovine	0.10 ± 0.02 ^ns^	0.12 ± 0.04 ^ns^	0.13 ± 0.04 ^ns^	0.12 ± 0.03 ^ns^		

IC_50_ and Trolox equivalent antioxidant capacity (TEAC) coefficients of hemoglobin bovine and human hydrolysates for the DPPH method. ns—not significant, * *p* > 0.05 (multiple *t*-tests of comparisons). ANOVA, Tukey, and GraphPad Prism.

**Table 4 ijms-24-13055-t004:** Hemoglobin bovine and human hydrolysates for the ABTS method.

Purification		1%	2%	8%	10%	NKT	TROLOX
IC_50_ (mg/mL)	Human	3.24 ± 0.38	3.56 ± 0.45 ^ns^	3.30 ± 0.65 ^ns^	3.26 ± 0.38 ^ns^	0.64 ± 0.06	0.58 ± 0.07
	Bovine	4.10 ± 0.56	3.74 ± 0.53	4.38 ± 0.56	3.88 ± 0.49	-	-
*TEAC*	Human	0.16 ± 0.05 ^ns^	0.16 ± 0.06 ^ns^	0.17 ± 0.04 ^ns^	0.17 ± 0.03 ^ns^	0.90 ± 0.05	1
	Bovine	0.14 ± 0.04	0.15 ± 0.06	0.13 ± 0.05	0.15 ± 0.04	-	-

IC_50_ and TEAC coefficients of hemoglobin bovine and human hydrolysates for the ABTS method. Comparisons between different IC_50_ of each percentage within the same condition showed no significant difference (ns) between IC_50_ of human and bovine of hemoglobin (multiple *t*-tests of comparisons; GraphPad Prism).

**Table 5 ijms-24-13055-t005:** Antimicrobial activity of human and bovine hemoglobin hydrolysate fractions.

	Hemoglobin Fractions (µg/mL)																	
*Bacteria strains*	1		2		3		4		5		6		7		8		9	
	B	H	B	H	B	H	B	H	B	H	B	H	B	H	B	H	B	H
*Kocuria rhizophila*	1	1	2	2	4	4	2	4	2	1	2	4	4	4	8	4	2	2
*Staphylococcus aureus*	2	2	4	4	4	8	2	4	2	2	2	4	8	-	-	-	8	4
*Listeria* *monocytogenes*	1	1	625	625	8	8	4	4	4	4	2	4	8	8	-	-	4	4
*Escherichia coli*	8	8	4	8	4	4	4	4	15.6	8	4	4	8	15.6	8	4	310	310
*Salmonelle Newport*	4	4	625	625	2	4	4	4	4	4	8	15.6	310	310	8	-	4	4
*Micrococcus luteus*	8	8	2	4	-	8	2	4	4	4	8	4	-	-	8	8	625	625

The minimum inhibitory concentration (MIC) of peptide hydrolysates was determined in a microtiter plate assay system after 24 h of incubation at 37 °C. Bovine (B)/human (H).

**Table 6 ijms-24-13055-t006:** Antimicrobial peptide sequences identified by UPLC-MS/MS from bovine and human hemoglobin hydrolysis fractions.

Fraction (min)		Position	Sequence	Molecular Weight (Da)	Bovine	Human
Antimicrobials	1 (6–8)	α137-141	TSKYR	653	+	+
	2 (8–10)	α99-105	KLLSHSL	796	+	KLLSHCL
		α100-105	LLSHSL	668	+	-
	3 (10–15)	α99-106	KLLSHSLL	910	+	KLLSHCLL
	4 (15–20)	α133-141	STVLTSKYR	1054	+	+
	6 (25–30)	α37-46	PTTKTYFPHF	1238	+	+
		α36-45	FPTTKTYFPH	1238	+	+
		α32-41	FLSFPTTKTY	1204	-	+
	7 (30–35)	α34-46	LSFPTTKTYFPHF	1585	+	-
		α33-46	FLSFPTTKTYFPHF	1732	+	-
Anti-oxydant		α137-141	TSKYR	653	+	+

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
