# Peer review of "Comparison of the Bioactive Properties of Human and Bovine Hemoglobin Hydrolysates Obtained by Enzymatic Hydrolysis: Antimicrobial and Antioxidant Potential of the Active Peptide α137-141"

_ijms, 2023, doi:10.3390/ijms241713055_

Round 1

Reviewer 1 Report

Please find my comments in the attached file.

Author Response

Reviewers 1

The manuscript “Obtaining of new antioxidant and antimicrobial peptides derived from human hemoglobin by peptide hydrolysis and comparison with these obtained by bovine hemoglobin» by Outman et al, is dealing with the enzymatic hydrolysis of human hemoglobin in order to try to identify a new bioactive peptide with possible antimicrobial and antioxidant properties. The main core of this manuscript is based on the comparison of the obtained data with the ones that are already published for bovine hemoglobin. The authors were not able to identify any new peptide sequence with potential antimicrobial and antioxidant properties and therefore, there is no novelty and high significance of the presented data. However, authors were able to demonstrate, for the first time, that human hemoglobin hydrolysates can also be used as potential antimicrobial and antioxidant derivatives, despite the fact that they do not elaborate the advantage of these hydrolysates over the bovine ones, which gives a certain scientific soundness of the manuscript. I would recommend acceptance after major revision and after addressing the below mentioned comments:

  1. The title of the manuscript “Obtaining of new antioxidant and antimicrobial peptides derived from human hemoglobin by peptide hydrolysis and comparison with these obtained by bovine hemoglobin” does not correspond to its content. There are no newly identified antioxidant and antimicrobial peptides derived from human hemoglobin enzymatic hydrolysis. If the authors refer to Table 6, the peptides presented there are already known, while those that differ in one amino acid (AA) composition were not separately tested if they possess the above-mentioned properties. As in the manuscript authors are discussing antimicrobial and antioxidant properties of the hydrolysates (that consist of lot of peptide sequences) they cannot automatically classify these peptides (the ones that differ in one AA) as antimicrobial and antioxidant without testing them separately.

Answer: Petide α137–141, TSKYR (neokyotorphin, NKT) a peptide derived from bovine hemoglobin, whose antimicrobial and antioxidant effects are already known in the scientific literature (Przybylski et al. 2016). It has also been found in porcine cruor with antimicrobial and antioxidant effects (zouari et al., 2020). In our study, this is the first time that this peptide has been identified in human hemoglobin (Figure 6C, fraction 1: Peptidomic analysis using UPLC-MS/MS and bioinformatics. Histogram showing the number of unique peptide sequences identified). In our first article published recently, we identified this peptide in particular in human hemoglobin with appropriate techniques (see article outman et al 2023). Here, this small α137–141, TSKYR

Zouari, O., Przybylski, R., Hannioui, M., Sion, L., Dhulster, P., & Nedjar-Arroume, N. (2020). High added-value co-product: The porcine cruor is an attractive source of active peptides. J. Nutr. Health Food Sci, 7, 1-9.

Outman, A.; Deracinois, B.; Flahaut, C.; Diab, M.A.; Gressier, B.; Eto, B.; Nedjar, N. Potential of Human Hemoglobin as a Source of Bioactive Peptides: Comparative Study of Enzymatic Hydrolysis with Bovine Hemoglobin and the Production of Active Peptide α137–141. Int. J. Mol. Sci. 2023, 24, 11921. https://doi.org/10.3390/ijms241511921

  1. Therefore, I completely disagree with the authors statement “This shows that this amino acid change does not lead to a total loss of antimicrobial activity [33]. “(page 11 lines 369-370), as it is known that single AA change in the sequence can influence dramatically on the peptide activity.

Answer: We had taken note of this advice and This statement has been deleted.

  1. Another error in the title is that it seems these new antioxidant and antimicrobial peptides are derived from human hemoglobin by peptide hydrolysis. The sequences described in Table 6 are derived by enzymatic hydrolysis of human hemoglobin, not by peptide hydrolysis. Overall, I strongly recommend the current title to be changed/rephrased in the next improved version of the manuscript.

Answer: We have taken note of this advice and we propose the following title.

Comparison of the bioactive properties of human and bovine hemoglobin hydrolysates obtained by enzymatic hydrolysis: antimicrobial and antioxidant potential of the active peptide α137-141.

  1. Page 1, lines 38-46, are not clear and very confusing. At one-point authors are talking about agrifood proteins, then suddenly move to discuss peptides and continuing with hemoglobin as representative of these proteins (it is totally unclear which proteins). Also, they are citing „Bah et al 2103, Castro et Sato 2015, Przybylski et al 2016 (lines 41-42) while in the line 45 they begin with citations „[1,2]“. Please be consistent regarding the citation style and numbering (similar examples can be found further in the text, e.g line 338).

Answer: We took note of this advice and made the corrections in the article.

  1. Lines 64-65, the sentence „However, human hemoglobin, like bovine hemoglobin, contains a large number of proteins that could be used to generate bioactive peptides. “Is incorrect. Hemoglobin does not contain a large number of proteins; it is a protein composed of 4 subunits.

Answer: We took note of this advice and made the corrections in the article.

  1. Lines 72-76, authors are mentioning that „The main objective of this study was to produce bioactive peptides through the enzymatic hydrolysis of human and bovine hemoglobin (used as a control group, laboratory model) and to demonstrate their bioactive properties. The focus was on the production of a specific peptide, α137- 141, which is often obtained through this method [13,15,20]. The secondary objectives were to evaluate the biological activity of the peptides. “What is the advantage of producing bioactive peptides by enzymatic hydrolysis of human hemoglobin over producing the same bioactive peptides by using bovine hemoglobin? Same question for the production of α137-141, why using human hemoglobin is more beneficiary than using bovine?

Answer: Human hemoglobin does indeed have particular advantages over bovine hemoglobin.

Advantages related to endogenous origin: these peptides offer the advantage of already being optimized by evolution to fulfill their specific functions in humans. This natural optimization increases the probability of discovering bioactive peptides that are both effective in their therapeutic actions and well tolerated by the human body. By exploiting human hemoglobin as a source of bioactive peptides, it is possible to benefit from their natural compatibility with the human biological system, which can facilitate the development of safer and more effective drugs targeting various diseases and conditions.

Reduced tolerance and immunogenicity: Derivatives of endogenous peptides, such as those derived from human hemoglobin, are generally better tolerated and less immunogenic than peptides based on foreign antigens. This may reduce the risk of adverse effects and intolerance when using these peptides for therapeutic purposes (Guaní-Guerra et al., 2010; Felício et al., 2017)

Take the example of cancer, which is one of the major public health problems and conventional treatments can cause many undesirable side effects. Indeed, several scientific studies have shown that certain antimicrobial peptides contained in human hemoglobin have demonstrated anti-cancer properties. In particular, selective cytotoxic effects on cancer cells while preserving normal cells (Guaní-Guerra et al., 2010; Gaspar et al., 2013; Matteo Bosso et al., 2018). By exploiting the specific properties of these peptides, it is possible to design more effective treatments with fewer unwanted side effects.

  1. Line 112, please change the word “peptides“ with „hydrolysates or „hemoglobin hydrolysates“. I notice that authors very often in the text refers to “peptides “, while more correct and precise term will be “hydrolysates“ or “hemoglobin hydrolysates“. Here are some examples (line 132, Table 2 caption; line 134; line 249; lines 543-544; line 631; line 634). Please check the text and correct this before submitting a new version.

Answer: We took note of this advice and made the corrections in the article.

  1. The numbers given in the Table 2 are missing corresponding units. Some of the numbers are with “.”, while some with “,”. Please be consistent. Part of the text written under the Table 2 should be included in the Table caption. Any explanation given under the Table should be correspondingly referenced (by *, 1, etc) and related with the data presented in the Table (this holds true for all Tables in the manuscript. In all cases text that is given under the Tables is not correspondingly related with the Table composition).

Answer: We took note of this advice and made the corrections in the article.

  1. Line 147, please give the whole name of BHT, you are using for the first time this abbreviation.

Answer: Corrections made

  1. Line 165, you are missing “8%”. Please include it.

Answer: Corrections made

  1. Line 189, “Ansignificativeincrease “, please correct this.

Answer: Corrections made

  1. Line 190, I think the values given in the brackets should be 1, 5 and 10.

Answer: Corrections made

  1. Line 198, I think that “substrate concentration“ is wrong and you refer to “hydrolysate concentration“.

Answer: Corrections made

  1. Line 205, I think “purification“ is wrong and should be replaced with „degree of hydrolysis“ (same in the line 208. Also here please replace „á” with a).

Answer: Corrections made

  1. Authors are mentioning that they are working with hydrolysates concentration of 1, 5 and 10 mg/mL, but they did not mention how these concentrations were calculated. Could you please explain how the final concentration of the hydrolysates was determined?

Answer: We had taken note of this advice and we add this information in the articles ref: 4.2.2 will now take it into account.

  1. Table 3 and 4, please replace “EC50” with “IC50”.

Answer: Corrections made

  1. Lines 222-228, the text is redundant, we already have all these data in Table 3. Please delete it.

Answer: Corrections made

  1. Line 251, you are missing author’s name before the reference 23.

Answer: Corrections made

  1. Line 265, please delete “CI 50” and replace it with “IC50”.

Answer: Corrections made

  1. Line 284, “highest” or “higher”?

Answer: Corrections made

  1. Lines 317-321 and Table 4, how do authors explain the differences in the antibacterial activity of fraction 1 vs antibacterial activity of whole hydrolysate? Namely, fraction 1 is showing much higher antibacterial activities (µg/mL) in comparison with the antibacterial activities of whole hydrolysates (that are 100 times less, mg/mL, lines 112-123)?

Answer: The differences in the antibacterial activity of fraction 1 compared to the whole hydrolyzate can be explained by the fractionation process which allowed the isolation of specific peptides with potentially higher antibacterial activities. While the whole hydrolyzate contains an array of peptides, some of which may interact or compete, which may reduce their antibacterial activity. However, the fractionation made it possible to collect different fractions at regular intervals, and fraction 1, in particular, showed significantly higher antibacterial activities (expressed in µg/mL) than those of the whole hydrolyzate (expressed in mg/ mL). The concentration and isolation of specific peptides, including the pure NKT peptide present in fraction 1, are well documented for their potent antibacterial potency in the scientific literature.

This fractionation work constitutes a first step towards the identification of specific antibacterial peptides. It is quite normal for certain fractions to show very high antibacterial activities compared to total hydrolysates, because fractionation allows the selection of specific peptides which may have more potent and targeted antibacterial properties. These results are encouraging and suggest that Fraction 1, in particular the pure NKT peptide (both human and bovine), could be a promising candidate for the development of future antibacterial agents.

  1. Lines 325-326, what do you mean by “more interesting”. Please elaborate.

Answer: Corrections made

  1. Line 334, Table 4 caption, what do you mean by “peptidomic”. It is not clear for me in which context peptidomic is used here.

Answer: Corrections made

  1. Line 355, figure 6 caption, what do you mean by “unique”. Please elaborate.

Answer: Corrections made

  1. Line 469, what is “solvent Bover 40”?

Answer: Corrections made

  1. In the Equation 1, you are missing “A blank”.

Answer: Corrections made

Reviewer 2 Report

The manuscript by Outman et al. reports the antioxidant and antimicrobial peptides derived from human hemoglobin and their comparison with peptides derived from bovine hemoglobin. The hydrolysates showed potent antimicrobial activity against several bacterial strains as well as antioxidant activity. Subsequently, new bioactive peptides were characterized by mass spectrometry to determine their amino acid sequence.

In my opinion this manuscript is of interest but some points need to be improved for its publication. Some information to be added or improved are listed below:

1. A short paragraph should be added giving information on the differences between the first structure of the 4 polypeptide chains of the two hemoglobins (similarity, identity, etc.) An additional figure with alignment would be necessary for the readers.

2. The authors state that they block the pepsin in 5 M NaCl and then freeze-dry the hydrolysates. This means that there was a high concentration of salt in the samples. Did they remove this excess NaCl? Or did they test equivalent saline solutions as a control when testing the hydrolysates on the bacteria they used?

3. What are the main primary structural differences between the peptides found in human and bovine hemoglobin? Is there any information in the literature about these bovine peptides? Only when structural evidence is provided can the authors claim to have characterised new biactive peptides.

Author Response

Reviewers 2

Comments and Suggestions for Authors

The manuscript by Outman et al. reports the antioxidant and antimicrobial peptides derived from human hemoglobin and their comparison with peptides derived from bovine hemoglobin. The hydrolysates showed potent antimicrobial activity against several bacterial strains as well as antioxidant activity. Subsequently, new bioactive peptides were characterized by mass spectrometry to determine their amino acid sequence.

In my opinion this manuscript is of interest, but some points need to be improved for its publication. Some information to be added or improved are listed below:

  1. A short paragraph should be added giving information on the differences between the first structure of the 4 polypeptide chains of the two hemoglobin (similarity, identity, etc.) An additional figure with alignment would be necessary for the readers.

Answer: We had taken note of this advice and we had added the information in the article.

  1. The authors state that they block the pepsin in 5 M NaCl and then freeze-dry the hydrolysates. This means that there was a high concentration of salt in the samples. Did they remove this excess NaCl? Or did they test equivalent saline solutions as a control when testing the hydrolysates on the bacteria they used?

Answer: Indeed, the action of pepsin has been neutralized by means of a 5M NaCl solution, administered in minimal quantities of the order of microliters, equivalent to one or two drops. Nevertheless, our team has engaged in a study aimed at exploring the hydrolysis of hemoglobin in order to produce bioactive peptides, by adopting an innovative and environmentally friendly approach: electrodialysis with bipolar membrane (EDBM). The objective of this approach was to identify the bioactive peptides generated through EDBM, while comparing them with those resulting from conventional hydrolysis, in order to assess their biological activity.

The hydrolysates obtained using EDBM, in a process free of chemical solvents and therefore without the addition of salt, revealed remarkable antimicrobial activity against six bacterial strains. They also demonstrated substantial antioxidant activity, confirmed by four separate tests, as well as efficacy against five strains of yeast and mold. The incidence of the presence or absence of salt does not seem to exert a significant influence on the biological activity of the hydrolysates. However, in view of ecological considerations, the adoption of a system such as EDBM could prove more appropriate for future stages of this research (Abou Diab et al., part I and II; 2020)

Abou-Diab, M.; Thibodeau, J.; Deracinois, B.; Flahaut, C.; Fliss, I.; Dhulster, P.; Nedjar, N.; Bazinet, L. Bovine Hemoglobin Enzymatic Hydrolysis by a New Ecoefficient Process—Part I: Feasibility of Electrodialysis with Bipolar Membrane and Production of Neokyotorphin (α137-141). Membranes 2020, 10, 257. https://doi.org/10.3390/membranes10100257

Abou-Diab, M.; Thibodeau, J.; Deracinois, B.; Flahaut, C.; Fliss, I.; Dhulster, P.; Bazinet, L.; Nedjar, N. Bovine Hemoglobin Enzymatic Hydrolysis by a New Eco-Efficient Process-Part II: Production of Bioactive Peptides. Membranes 2020, 10 , 268. https://doi.org/10.3390/membranes10100268

  1. What are the main primary structural differences between the peptides found in human and bovine hemoglobin? Is there any information in the literature about these bovine peptides? Only when structural evidence is provided can the authors claim to have characterized new bioactive peptides.

Answer: Mass spectrometric analysis after 3 hours of enzymatic hydrolysis by pepsin identified a list of peptides from both bovine and human hemoglobin. Peptides derived from the hydrolysis of bovine hemoglobin are well documented and some of these peptides, which are claimed to be new to human hemoglobin, are actually identical without any amino acid changes. Therefore, we can conclude that these peptides share the same activities as those already studied in bovine. For example, our target peptide, TSKYR, was found identically in human hemoglobin. This peptide has already demonstrated antimicrobial and antioxidant activities by our team.

Reviewer 3 Report

It has to be demonstrated that the hemoglobin peptides are devoid of hemolytic activity as an indication of nontoxicity.

 Inclusion of positive controls in the assays for comparison with the hemoglobin peptides would be desirable.

Consider performing assays of antifungal and antiviral activities on the hemoglobin peptides.

How stable are the peptides e.g. to peptidases?

Table 6 bottom line:"antioxidant "is misspelled

section 2.12.1 line 2:hydrolysates,NOT hydrolysis

Line 8 below Table 4: no a significance(p<0.005) looks contradictory

Some language polishing is necessary

Author Response

Reviewers 3

Comments and Suggestions for Authors

  1. It has to be demonstrated that the hemoglobin peptides are devoid of hemolytic activity as an indication of nontoxicity.

Answer: Indeed, studies conducted within the laboratory revealed that none of the antimicrobial peptides caused hemolysis of bovine red blood cells, even at high concentrations reaching five times the Minimum Inhibitory Concentration (MIC). These results demonstrate the fact that hemoglobin-derived peptides do not exhibit toxicity towards red blood cells. This observation is in agreement with the conclusions of the thesis entitled "Preparation of antimicrobial peptides from the enzymatic hydrolysis of two proteins: bovine hemoglobin and bovine α-lactalbumin" written by Véronique Dubois.

  1. Inclusion of positive controls in the assays for comparison with the hemoglobin peptides would be desirable.

Answer: We thank you for this advice and we will make good use of it in our future studies as you have suggested.

  1. Consider performing assays of antifungal and antiviral activities on the hemoglobin peptides.

Answer: We thank you for this advice and we will make good use of it in our future studies as you have suggested.

How stable are the peptides e.g. to peptidases?

Answer: The stability of hemoglobin-derived peptides against peptidases is highly dependent on environmental conditions. Pepsin, an enzyme present in the cells of the gastric mucosa, plays a key role in the hydrolysis of peptides. Its catalytic efficiency is maximum under acidic conditions similar to those of gastric juice, which is released in the stomach of vertebrates. Its optimum activity is generally in a pH range of 1 to 6, with an optimum value of around 3.5. The optimum temperature range for its activity is between 37 and 42°C, corresponding to the hydrolysis conditions used in this study.

It is important to note that commercial pepsin, such as that used in this research (Sigma; P6887), exhibits an activity generally between 3200 and 4500 units per milligram of protein upon receipt, and its content is regularly evaluated according to a specific protocol.

Following the completion of enzymatic hydrolysis, the abrupt cessation of the action of pepsin is achieved by adjusting the pH to 9. This ensures the inactivation of pepsin, since its activity is lost in an alkaline medium, usually at a pH greater than 8.5, due to its denaturation.

To avoid any undesirable further hydrolysis, the samples are immediately stored at -20°C. Although advanced analytical techniques such as high-performance liquid chromatography (HPLC) and mass spectrometry were used to monitor peptide concentrations over time, no significant changes were observed. However, it should be noted that the tests on the hemoglobin hydrolysates were carried out shortly after their production, thus minimizing the influences of time on the stability of the peptides.

Table 6 bottom line: «antioxidant "is misspelled.

Answer: Corrections made

Section 2.12.1 line 2: hydrolysates, NOT hydrolysis

Answer: Corrections made

Line 8 below Table 4: no a significance (p<0.005) looks contradictory

Answer: Corrections made

Round 2

Reviewer 2 Report

The authors have partially responded to requests that the revised manuscript may be accepted for publication if the editor agrees.

Reviewer 3 Report

The revised manuscript can be accepted for publication